# Quantifying electron-transfer in liquid-solid contact electrification and the formation of electric double-layer

Shiquan Lin[1,2], Liang Xu [1,2], Aurelia Chi Wang[3] & Zhong Lin Wang [1,2,3]*

Contact electrification (CE) has been known for more than 2600 years but the nature of charge carriers and their transfer mechanisms still remain poorly understood, especially for the cases of liquid–solid CE. Here, we study the CE between liquids and solids and investigate the decay of CE charges on the solid surfaces after liquid–solid CE at different thermal conditions. The contribution of electron transfer is distinguished from that of ion transfer on the charged surfaces by using the theory of electron thermionic emission. Our study shows that there are both electron transfer and ion transfer in the liquid–solid CE. We reveal that solutes in the solution, pH value of the solution and the hydrophilicity of the solid affect the ratio of electron transfers to ion transfers. Further, we propose a two-step model of electron or/and ion transfer and demonstrate the formation of electric double-layer in liquid–solid CE.

---

[1] Beijing Institute of Nanoenergy and Nanosystems, Chinese Academy of Sciences, Beijing 100083, PR China. [2] School of Nanoscience and Technology, University of Chinese Academy of Sciences, Beijing 100049, PR China. [3] School of Materials Science and Engineering, Georgia Institute of Technology, Atlanta, GA 30332-0245, USA. *email: zlwang@gatech.edu

Contact electrification (CE) (or triboelectrification in general terms) is a universal but complicated phenomenon, which has been known for more than 2600 years. The solid-solid CE has been studied using various methods and different mechanisms were proposed (Electron transfer[1,2], ion transfer[3] and material transfer[4–6] were used to explain different types of CE phenomena for various materials). In parallel, CE between liquid–solid is rather ubiquitous in our daily life, such as flowing water out of a pipe is charged, which is now the basis of many technologies and physical chemical phenomena, such as the liquid–solid triboelectric nanogenerators (TENGs)[7–10], hydrophobic and hydrophilic surfaces, and the formation of electric double-layer (EDL)[11–14]. However, understanding about the liquid–solid CE is rather limited and the origin about the formation of EDL remains ambiguous owing to the lacking of fundamental understanding about charge transfer at interfaces. The most important issue in the CE mechanism is the identity of charge carriers (electrons or/and ions), which has been debated for decades in the solid-solid CE[15,16]. Most recently, charge carriers have been identified as electrons for solid-solid CE based on temperature dependent effect and photoexcitation effect on the charged surfaces, and the ion transfer is out of consideration[17–19].

As for the case of liquid–solid CE, it is usually assumed to be ion transfer without any detailed studies, simply because ions are often present in liquids, such as H[+] and OH[−] in water. Regarding the nature of EDL, the charging of the isolated surfaces in a liquid is considered to be induced by ionization or dissociation of surface groups and the adsorption or binding of ions from liquid onto the solid surface[20]. From these points of view, the charge carriers in liquid–solid CE is naturally assumed to be ions and transfer of electrons has not been even considered. However, Wang et al. has proposed a "electron-cloud-potential-well" model for explaining CE in a general case, in which the electron transfer in CE is considered to be induced by the overlap of electron clouds as a result of mechanically forced contact[18]. At a liquid–solid interface, the molecules in a liquid collide with atoms on the solid surface owing to liquid pressure, which may lead to the overlap of electron clouds and result in electron transfer. Hence, there is still dispute about the identity of charge carriers in the liquid–solid CE, which is one of the most fundamental questions in CE and physical chemistry as well. Such a question can now be answered using the surface charge decay experiments at different temperatures for distinguishing electron transfer from ion transfer in liquid–solid contact[17,18]. This is because electrons are easily emitted from the solid surface as induced by thermionic emission, while ions usually bind with the atoms on the solid surface, and they are rather hard to be removed from the surface in comparison to electrons especially when the temperature is not too high.

Here we show the CE in liquid–solid and the charge density on solid surfaces after the contact measured using Kelvin probe force microscopy (KPFM)[21–24]. We investigate the decay of CE charges on the solid surfaces at different temperatures. We particularly study the effects of solutes in the aqueous solution, pH value of the aqueous solution and the hydrophilicity of the solid surfaces on the liquid–solid CE. We have analyzed the ratio of electron transfers to ion transfers in the liquid–solid CE for the first time according to the thermionic emission theory, to the best of our knowledge. Lastly, we propose a model about the formation of the EDL based on the understanding of the charge transfer at liquid–solid interface, providing a distinct mechanism from the general understanding in classical physical chemistry.

## Results

**The CE between the DI water and the SiO$_2$.** Here, flat insulating ceramic thin films, such as SiO$_2$, Al$_2$O$_3$, MgO, Ta$_2$O$_5$, HfO$_2$, AlN,

and Si$_3$N$_4$., deposited on highly doped silicon wafers, were used as solid samples. The liquids were chosen as deionized water (DI water) and different aqueous solutions, including NaCl, HCl and NaOH solutions. In the experiments, the liquid dropped from a grounded needle and slid across the ceramic surface, as shown in Fig. 1a. After the liquid being vaporized, the surface charge densities on the ceramic surfaces were measured by using KPFM at different substrate temperatures. According to previous studies, ions will be produced by ionization reaction on the oxide and nitride surfaces when they contact the aqueous solutions[25–29]. For example, O[−] ions will be generated by the ionization reaction on the SiO$_2$ surface as shown below (The hydroxyl on the SiO$_2$ surface is usually produced by adsorbing water molecules in the air or contacting with water)[29]:

$$\equiv Si - OH + OH^- \Leftrightarrow \equiv Si - O^- + H_2O \qquad (1)$$

As introduced above, electrons may be another type of charge carrier on the SiO$_2$ surface after contacting with aqueous solutions. Hence, we assume that there are both O[−] ions and electrons on the liquid sliding trace on the SiO$_2$ surface, as shown in the inset in the Fig. 1a. When the SiO$_2$ sample is heated by the sample heater, the electrons will be thermally excited and emitted from the surface, as shown in Fig. 1b, while the O[−] ions may stay on the surface since they formed covalent bonds with the Si atoms on the SiO$_2$ surface. (As shown in the *ab initio* molecular dynamics simulations in the Supplementary Note 1, Supplementary Figs. 1, 2, and the simulation results are shown in Supplementary Movies 1–7). This means that, if heating can induce obvious decay of CE charges on the SiO$_2$ surface, it may be mainly caused by thermal emission of electrons.

In the experiments, the CE between the SiO$_2$ and the DI water was first performed, and Fig. 1c gives the results of the temperature effect on the decay of CE charges on the SiO$_2$ surfaces. It is obvious that the SiO$_2$ is negatively charged and the charge density on the SiO$_2$ surface is about $-810 \, \mu Cm^{-2}$ (negative sign means that the charges are negative) after the contact with the DI water. In Fig. 1c, the temperature affects the decay of the negative charges on the SiO$_2$ surface significantly. The surface charge density on SiO$_2$ remains almost unchanged at 313 K and slight decay of the surface charge density is observed at 343 K. As the sample temperature continues to rise, the decay rate of the surface charges increases. But some charges (about $-180 \, \mu Cm^{-2}$) cannot be removed even when the temperature rises up to 434 K and 473 K (these charges can be called as "sticky" charges that remain on surfaces even when the temperature is raised). For the removable charges, the decay behaviors are consistent with the thermionic emission theory, in which the electrons are considered to obtain more energy and the electron density decay faster at higher temperatures. Moreover, it is found that the charge density decay exponentially and the decay curve follows the electron thermionic emission model as described by the following equation, which was proposed in our previous studies (The curve fitting results are shown in Supplementary Fig. 3)[17,18]. Hence, the removable charges in the CE between the SiO$_2$ and the DI water can be identified as electrons.

$$\sigma = e^{-at}\sigma_e + \sigma_s \qquad (2)$$

where $\sigma$ denotes the CE charge density on the sample surface, $\sigma_e$ denotes the initial density of charges on the sample surface, which can be removed by thermal excitation, $\sigma_s$ denotes the density of the "sticky" charges, which cannot be removed by heating and $t$ denotes the decay time.

For the "sticky" charges, charging and heating cycle tests were performed to observe their behaviors, as shown in Fig. 1d. In

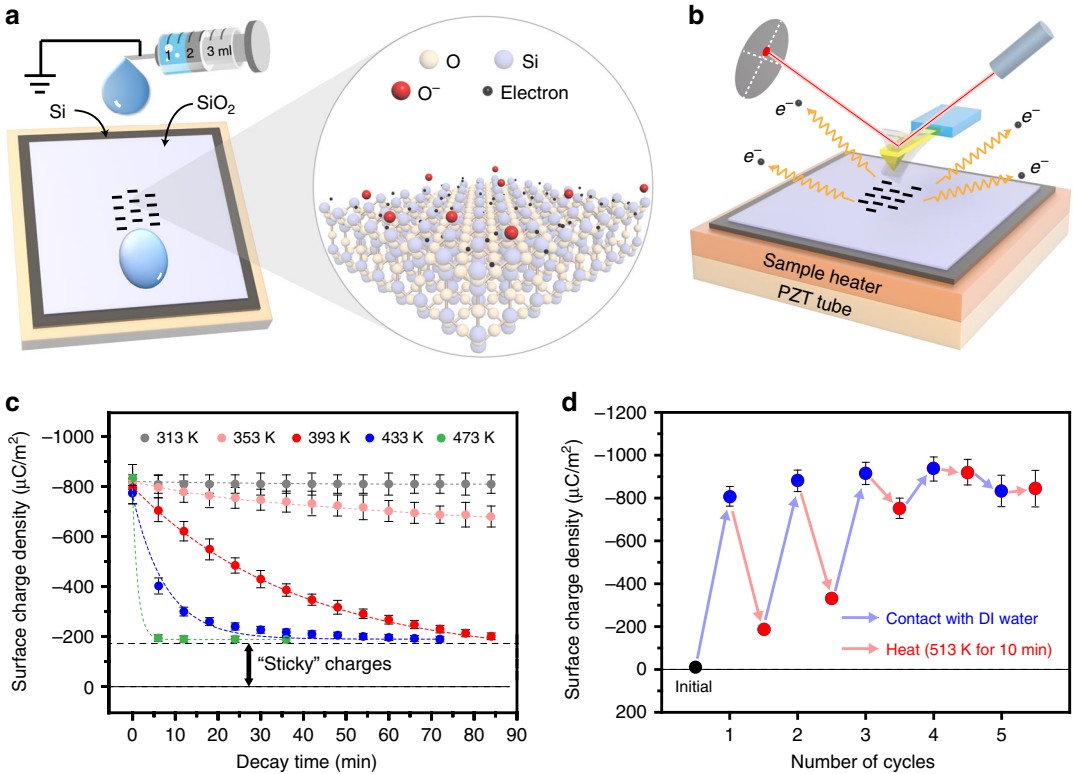

**Fig. 1 Temperature effect on the CE between the DI water and the SiO$_2$.** (**a**) The setup of the charging experiments, where the negative charges generated on the SiO$_2$ surface could be electrons and O$^-$ ions induced by surface ionization reaction. ('O' is the Oxygen atom, 'Si' is the silicon atom and 'O$^-$' is the Oxygen ion). (**b**) The setup of AFM platform for the thermionic emission experiments. (**c**) The decay of the CE charge (induced by contacting with the DI water at room temperature) on the SiO$_2$ surface at different substrate temperatures. (**d**) The CE charge density on the SiO$_2$ sample surface in the charging (contacting with the DI water at room temperature) and heating (at 513 K for 10 min) cycle tests. (Error bar are defined as s. d.).

every cycle of the testes, the SiO$_2$ sample contacts with the DI water first, and then it is heated to 513 K and maintains for 10 min to remove the electrons on the surface. In the first cycle, the SiO$_2$ is negatively charged when it contacts with the DI water, and the density of the "sticky" charges is −180 μCm$^{-2}$ as expected. It is found that the density of the "sticky" charges increases to −300 μCm$^{-2}$ in the second cycle and it continuously increases with the number of the cycles. After five cycles of experiments, the density of the "sticky" charges reaches a saturation value, and there are not removable charges on the SiO$_2$ surface. These behaviors suggest that the "sticky" charges should be ions, such as O$^-$ ions, instead of electrons. As shown in Supplementary Fig. 4, in each contact with the DI water, both electrons and O$^-$ ions are attached on the surface. Electrons are emitted as temperature rises, while the O$^-$ ions cannot be removed in the subsequent heating if the temperature is not too high. In the next cycle of introducing water droplet, more O$^-$ ions are produced in the ionization reaction and accumulate on the SiO$_2$ surface since it has not reached saturation. With the increase of cycles on introducing water droplets, the concentration of O$^-$ ions continues to rise and more "available charge positions and densities" are filled, thus, it becomes harder for the SiO$_2$ to gain more electrons in the CE, resulting in a decrease of the electron density on the surface. A few cycles later, the density of the ions reaches a saturated value, which remains stable even in the followed heating process.

Based on the analysis, it turns out that electrons can be distinguished from ions in the CE by performing the thermionic emission experiments. The removable and the "sticky" charges in the experiments are identified as electrons and ions, respectively. And the results suggest that there are both electron and ion

transfers in the CE between the SiO$_2$ and the DI water. The density of transferred electrons is measured to be −630 μCm$^{-2}$ and the density of transferred ions is about −180 μCm$^{-2}$. It means that the electron transfer, which account for 77% of the total charges, is dominant in the CE between SiO$_2$ and DI water in very first contact.

**Effects of the solutes and the liquid pH value on the CE.** Further, the effects of the solutes in the liquid and pH value of the liquid on the liquid–solid CE were studied. The CE between the SiO$_2$ and different aqueous solutions, including NaCl, HCl and NaOH solutions, was performed and the electron transfer and ion transfer in the CE were separated by the heat-induced charge decay experiments. Figure 2a gives the effect of the NaCl concentration on the transferred charge density in the CE between the SiO$_2$ and the NaCl solution. It is found that the charge density on the SiO$_2$ surfaces decreases with the increase of the NaCl concentration. This result is consistent with the previous studies about the liquid–solid TENG, in which the salt solution is the liquid and the output of the TENG decreases with the increase of the salt concentration[30,31]. The effect was not clearly explained before, because there was no method to identify the charge carriers. Here, the decay of the charge density on the SiO$_2$ surfaces is performed at 433 K after the CE between the SiO$_2$ and the NaCl solutions, and the results are shown in Fig. 2b. It can be seen that the charge density decays exponentially, which is the same as the CE between the SiO$_2$ and the DI water as introduced above. The density of removable charges (electrons) on the SiO$_2$ surfaces decreases with the increase of the NaCl concentration, while the "sticky" charges (ions) density remains almost unchanged when

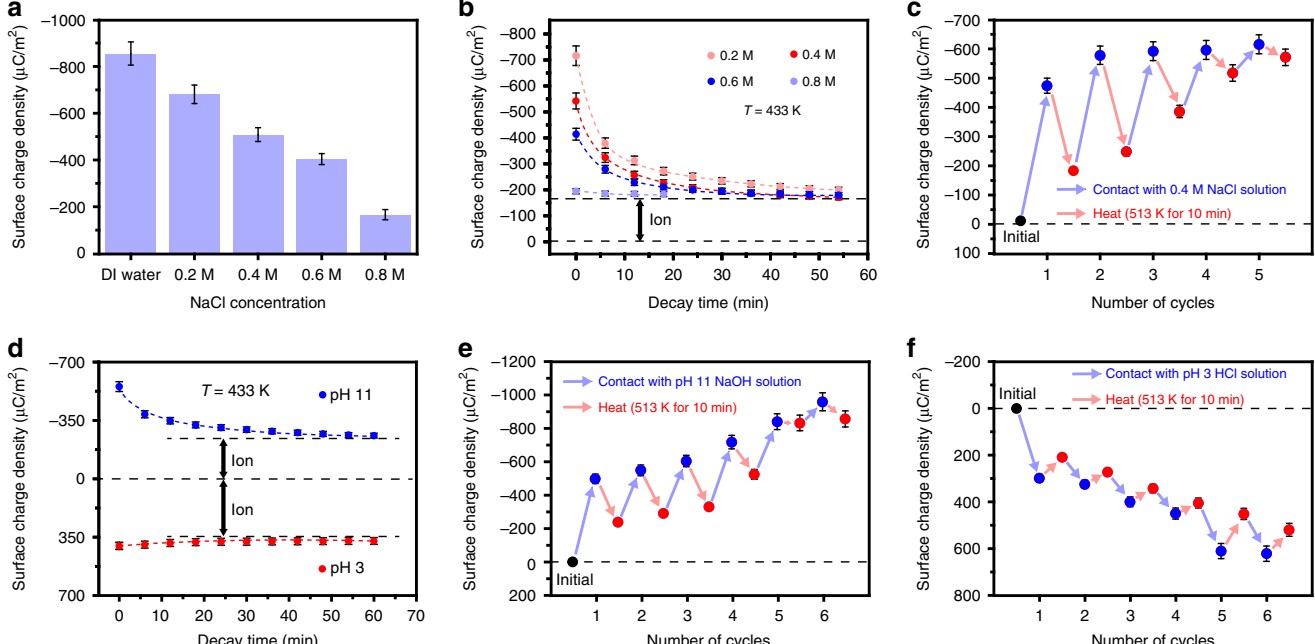

**Fig. 2 Temperature effect on the CE between the SiO₂ and aqueous solutions.** (**a**) The effects of the NaCl concentration on the CE between the SiO₂ and the NaCl solutions. (**b**) The decay of the CE charge at 433 K which is induced by contacting with the NaCl solutions. (**c**) The CE charge density on the SiO₂ sample surface in the charging (contacting with 0.4 M NaCl solution at room temperature) and heating (513 K for 10 min) cycle tests. (**d**) The decay of the CE charge at 433 K, which is induced by contacting with the pH 11 HCl solution and the pH 3 NaOH solution. The charging and heating cycle testes when the liquids are (**e**) the pH 11 NaOH solution and (**f**) the pH 3 HCl solution. (Error bar are defined as s. d.).

the SiO₂ contacts with the NaCl solutions of different concentrations. It implies that the decrease of the charge density on the SiO₂ induced by the increase of the NaCl concentration is mainly due to the decrease of the electron transfer, which might be caused by the increase of the dielectric constant of the NaCl solution that facilitate the discharge after charging. Different from electron transfer, the ion transfer will not be significantly affected by the NaCl concentration in the first contact (Fig. 2b). This result is easy to understand, because there are no Na⁺ or Cl⁻ in the ionization reaction (chemical formula 1), which produce the required O⁻ ions on the SiO₂ surface. Figure 2c gives the CE charge density on the SiO₂ sample surface in the charging (contacting with 0.4 M NaCl solution) and heating (513 K for 10 min) cycle tests. The results show that the saturated ion density in the CE between 0.4 M NaCl solution and SiO₂ is slightly lower than that between DI water and SiO₂ (Fig. 1d). The difference in the saturated ion density may be caused by the covering of the crystallized NaCl on the SiO₂ surface in the subsequent heating processes, which blocks the progress of ionization reaction.

Different from Na⁺ or Cl⁻, it can be seen that the OH⁻ plays an important role in the generation of the O⁻ ions on the SiO₂ surface from the chemical formula 1. Hence, the density of the transferred ions on the SiO₂ surface may be affected by the pH value of the solutions. Figure 2d shows the decay of the surface charge density on the SiO₂ surface, which is produced by contacting with the pH 11 NaOH solution and pH 3 HCl solution. When the pH value of the solution increases to 11, the electron transfers decrease, and the density of transferred ions (about −230 μCm⁻²) is slightly higher than that when the pH value of liquid is 7 (DI water). And the difference can also be observed in the charging and heating tests, in which the saturated ion density on the SiO₂ surface when the liquid is the pH 11 NaOH solution is higher than that when the liquid is the DI water, as shown in Fig. 2e. This is caused by the increase of the OH⁻ concentration in the solution, which promotes the

ionization reaction (chemical formula 1). When the pH value of the solution changes to 3, the electron transfer direction and the polarity of the transferred ions on the SiO₂ surface reverse from negative to positive (Fig. 2d). And the charging and heating cycle tests in Fig. 2 f show that the positive ions also accumulate on the SiO₂ surface. In this case, the positive ions on the SiO₂ surface are produced by another ionization reaction, as shown below[26–28].

$$\equiv Si - OH + H^+ \Leftrightarrow \equiv Si - OH_2^+ \qquad (3)$$

The effects of pH value on the CE between liquids and various ceramics are shown in Supplementary Fig. 5. The results are similar to the pH effects on the SiO₂ surface. When the pH value of the solution was 11, the transferred ions on the ceramic surfaces are negative as shown in Supplementary Fig. 5a–c. When the pH value of the solution changes to 3, the polarity of the transferred ions also reverses to be positive as shown in Supplementary Fig. 5d–f. This means that the effects of pH value on the ionization reaction for different ceramics are consistent.

These results show that no matter what the aqueous solution is, there are always both electron transfer and ion transfer in liquid–solid CE. The electron transfers between aqueous solution and solid is sensitive to solutes in the liquids, such as Na⁺, Cl⁻, OH⁻ and H⁺ etc. While the ion transfer is mainly affected by the pH value of the solution, which dominates the ionization reactions on the insulator surfaces.

**Solid effects on the liquid–solid CE.** As another side in the liquid–solid CE, different solids were also tested in the thermionic emission experiments. As shown in Fig. 3a–f, the CE charge decay in the CE between the DI water and different insulating ceramics was performed, including MgO, Si₃N₄, Ta₂O₅, HfO₂, Al₂O₃ and AlN. (The surface ionization reaction equations between water molecules and these materials are shown in Supplementary

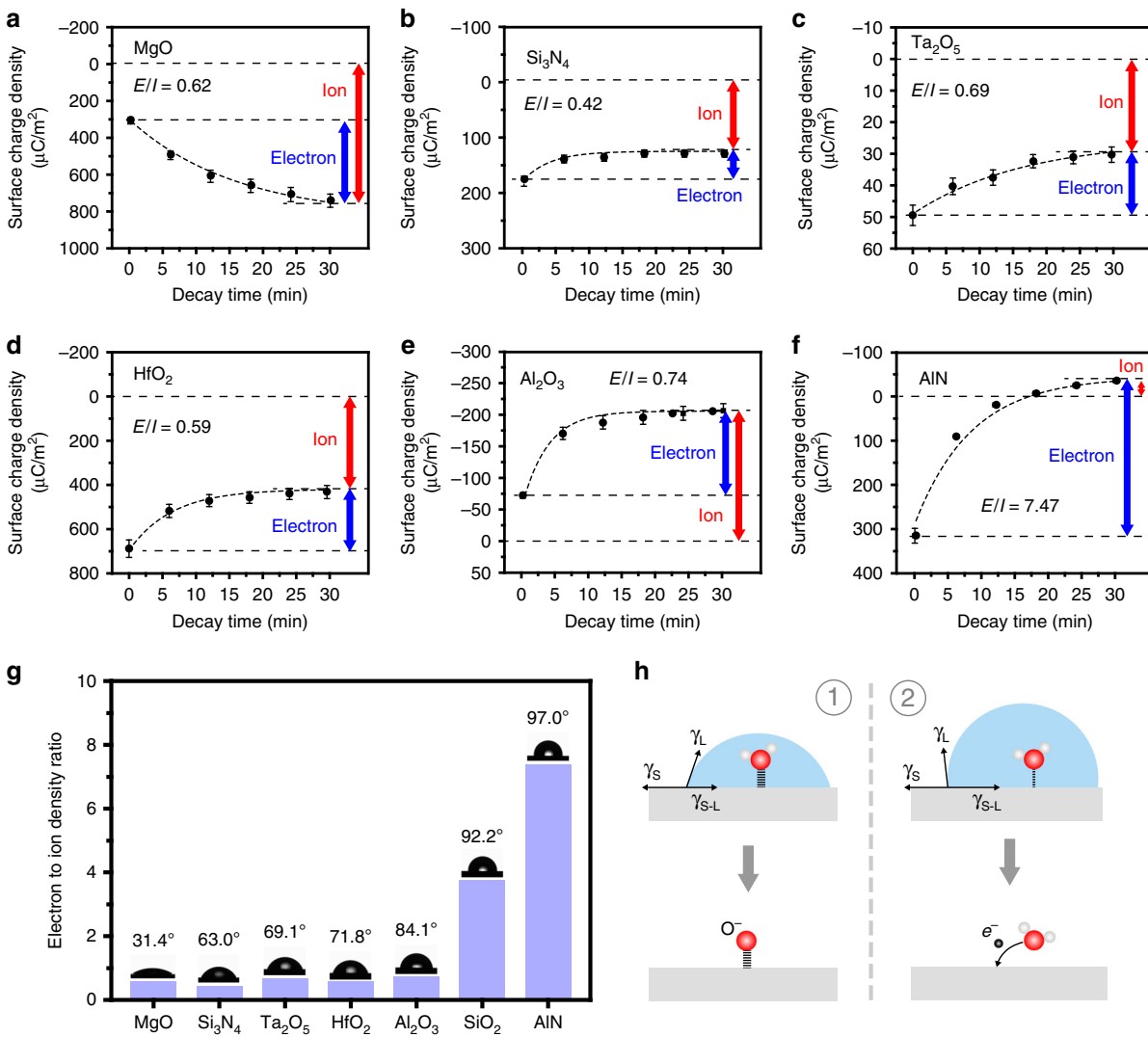

**Fig. 3 Temperature effect on the CE between the DI water and the solids.** The decay of CE charges (induced by contacting with the DI water at room temperature) on **a** MgO, **b** $Si_3N_4$, **c** $Ta_2O_5$, **d** $HfO_2$, **e** $Al_2O_3$, and **f** AlN surfaces at 433 K, and the amount of the electron transfer and the ion transfer in the CE between the DI water and different insulators. **g** The relation between the electron transfer to the ion transfer ratio and the DI water contact angle (WCA) of the materials. **h** The schematic of WCA effects on the ion transfer and electron transfer in liquid–solid CE. $\gamma_L$, $\gamma_S$, and $\gamma_{L-S}$ denote the liquid–gas interfacial tension, solid–gas interfacial tension and liquid–solid interfacial tension, respectively. (Error bar are defined as s. d.).

Note 2). It is found that all of the charge decay curves follow the electron thermionic emission model, hence the removable charges are electrons and the "sticky" charges are ions as analyzed above. The electron transfer and the ion transfer are marked in Fig. 3a–f, it can be seen that the ratio of electron transfers to ion transfers (E/I) highly depends on the type of solid. For the CE between the AlN and the DI water, more than 88% of the total transferred charges are electrons. But in the CE between the $Si_3N_4$ and the DI water, electron transfer is only 31% of the total charge transfer. In order to test the interaction between a liquid and a solid at the interface, the water contact angle (WCA) of the ceramics was measured and the results are shown in Fig. 3g. It is noticed that the E/I ratio slightly increases with the increase of the WCA when the WCA of materials is less than 90°. When the WCA of the materials increase to be larger than 90°, such as 92.2° for the $SiO_2$ and 97.0° for the AlN, the E/I ratio increases rapidly. For the $SiO_2$ and the AlN, the E/I ratios are 3.5 and 7.5, respectively. Actually, the WCA is dependent on the liquid–solid, solid-gas, liquid-gas interfacial tensions, which are related to the interfacial energy of two phases, as shown in the Fig. 3h. The interfacial energy

between a hydrophilic surface (WCA < 90°) and water is usually lager than that between a hydrophobic surface (WCA > 90°) and water. It means that the interaction between the water molecules and the solid surface with small WCA is usually stronger than that between water molecules and the solid surface with large WCA. And the Oxygen atoms or Hydrogen atoms in water molecules are more likely to form covalent bonds with the atoms on the hydrophilic surface. In other words, the surface ionization reaction is more likely to occur and leading to the generation of ions on the hydrophilic solid surface. On the contrary, the surface ionization reaction between the hydrophobic solid surfaces and water is less likely to occur, and the CE between the solid and aqueous solution is electron-dominated.

It needs to be mentioned that the polarity of the transferred electrons and transferred ions not necessary to be the same in liquid–solid CE. As shown in Fig. 3a, the MgO obtains electrons and positive ions at the same time in the CE between MgO and DI water (Supplementary Fig. 6a), and the positive charge density on the MgO surface increases in the heating due to the emission of electrons. For the CE between AlN and DI water, the AlN loses electrons and

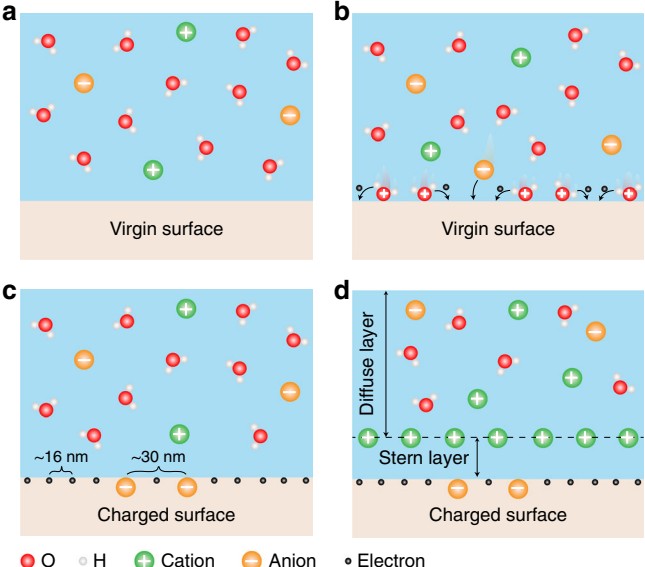

**Fig. 4 Mechanism of liquid–solid CE and formation of electric double-layer.** **a** The liquid contacts a virgin surface (before CE). **b** The water molecules and ions in the liquid impact the virgin surface and electron transfer between the water and the surface. **c** The surface is charged and the charge carriers are mainly electrons (WCA > 90°, pH = 7), some ions may be generated on the surface caused by the ionization reaction etc. **d** The opposite polarity ions are attracted to migrate toward the charged surface by the Coulomb force, electrically screening the first charged layer.

obtains negative ions (Supplementary Fig. 6b). These results suggest that the electron transfer and ion transfer in liquid–solid CE are independent of each other. Furthermore, it may be possible that the electron transfer and ion transfer could be segregated on different surface areas, but remain proved experimentally.

According to the results, the CE between solid and liquid can be affected by the pH value of the aqueous solution, solutes in the aqueous solution and the hydrophilicity of the solids. Nevertheless, there is always electron transfer in the CE between liquid (aqueous solution) and solid. This result was predicted in the "two-step" model first proposed by Wang et al.[32], but was not included at all in the classical explanation regarding the formation of the EDL. Combining the experiment results and the "two-step" model[32], a new picture for the liquid–solid CE and the formation of the EDL is proposed, as shown in Fig. 4. In the first step, the liquid contacts a virgin solid surface (Fig. 4a), the molecules and ions, including $H_2O$, cation, anion etc., will impact the solid surface due to the thermal motion and the pressure from the liquid (Fig. 4b). During the impact, electrons will transfer between the solid atoms and water molecules owing to the overlap of the electron clouds of the solid atoms and water molecules[18], and the ionization reaction may also occur simultaneously on the solid surface. Hence there will be both electrons and ions generated on the surface. As an example, the electron transfer plays a dominated role in the CE between the $SiO_2$ and DI water, as shown in Fig. 4c. In the second step, the opposite ions in the liquid would be attracted to migrate toward the charged surface by the electrostatic interactions, forming an EDL, as shown in Fig. 4d.

An atom with extra/deficient electrons are referred as ion, therefore, the transferred electrons on the solid surface is considered as the first step to make the "neutral" atoms on solid surface become ions in the "two-step" model[32]. From this perspective, the ions produced by the ionization reaction in the experiments can also be considered as the "neutral" atoms with extra electrons. The difference is that the transferred electrons

directly induced by the collisions between the atoms in the liquid and the atoms on the solid surface were usually trapped in the surface states, while the extra electrons of the "neutral" atoms produced in the ionization reaction were trapped in the atomic orbitals of the atoms (the atomic orbitals can be considered as the special surface states of solids generated in the ionization reaction). There is no essential difference between the electrons in the surface states and those in the atomic orbitals. However, the potential barrier of the surface states to prevent the electrons from emitting in the heating process might be lower than that of atomic orbitals. Hence, the electrons in the surface states of the solid are removable, while the electrons in the atomic orbitals are tightly bonded on the solid surfaces.

Also, the surface charge density (electrons and ions) in the liquid–solid CE is not as dense as that appearing in text book drawing. The highest transferred electron density in our experiments is $-630\ \mu C\ m^{-2}$ in the CE between $SiO_2$ and DI water, which corresponds to ~1 excess electron per 250 $nm^2$. Thus, the probability of electron transfer in liquid–solid CE is usually less than one out of ~2500 surface atoms. The transferred ion density in CE between $SiO_2$ and DI water is $-180\ \mu C\ m^{-2}$, which corresponds to ~1 $O^-$ ion per 1000 $nm^2$. Accordingly, the distance between two adjacent electrons on $SiO_2$ surface is ~16 nm, and the distance between two adjacent $O^-$ ions is ~30 nm, as shown in Fig. 4c. These distances are much larger than the thickness of Stern layer, which is of the order of a few ångströms[20]. Hence, the distance of two adjacent charges (electrons or/and ions) should be considered in the structure of the EDL.

## Discussion

In conclusion, the CE between liquid and solid was performed and the temperature effect on the decay of the CE charge on the ceramic surfaces was investigated. It is revealed that there are both electron transfer and ion transfer in the liquid–solid CE. The results suggest that the solutes in the aqueous solution, such as $Na^+$ and $Cl^-$ etc., can reduce the electron transfer between aqueous solution and solid. And the ion transfers in the liquid–solid CE induced by the ionization reaction can be significantly affected by the pH value of the liquid. Besides, it is found that the CE between hydrophilic surfaces and aqueous solutions is likely dominated by ion transfer; and the CE between hydrophobic surfaces and aqueous solutions is more likely to be dominated by electron transfer. This is the first time that the "two-step" model about the formation of EDL, in which the electron transfer plays a dominant role in liquid–solid CE, is verified experimentally. Our results may have great implications in the studies of TENG and EDL.

## Methods

**Sample preparation.** The $SiO_2$ layer was deposited on high doped silicon wafer by thermal oxidation. The $Si_3N_4$, $Al_2O_3$, $Ta_2O_5$, MgO, $HfO_2$, AlN layers were deposited on high doped silicon surfaces by magnetron sputtering, and the thickness of all the layers were 100 nm. The DI water with a resistivity of 18.2 MΩ cm used here was produced by deionizer (HHitech, China). Before the experiments, all the samples were heated for 10 min at 513 K to remove the charge on the surfaces. After the heat treatment, the charge density of the ceramic surfaces was measured to be about 0 $\mu C\ m^{-2}$, except the MgO and $Si_3N_4$. The "sticky" charge density on the MgO and $Si_3N_4$ surface was about 800 $\mu C\ m^{-2}$ and $-250\ \mu C\ m^{-2}$ before the CE with solutions, respectively. The "sticky" charges on the MgO and $Si_3N_4$ surfaces may be the ions generated by the ionization reaction between the samples and the water molecules in the air, since the MgO and $Si_3N_4$ are most hydrophilic in these ceramics.

**KPFM experiments.** The experiments were performed on commercial AFM equipment Multimode 8 (Bruker, USA). NSC 18 (MikroMash, USA; Au coated; tip radius: 25 nm; spring constant: 2.8 $Nm^{-1}$) was used as the conductive tip here. The sample temperature was controlled by the sample heater and the tip temperature was controlled by the tip heater independently. In all the experiments, the temperature of the sample and the tip remained consistent. The tapping amplitude was 350 mV, the scan size was 5 µm and the lift height was set to 50 nm in the KPFM

measurements. In order to acquire the data from a big region, the KPFM was manual operated to scan different positions on the whole sample surface (>20 positions). All the heating and charge measurements are performed in an Ar atmosphere. The changes of surface charge density were demonstrated not caused by the adsorption and desorption of the water molecules on $SiO_2$ surface, as shown in Supplementary Note 3 and Supplementary Fig. 7. And the observed changes in the surface potential in our experiments were not due to the temperature effects on the measurements, as shown in Supplementary Note 4 and Supplementary Fig. 8.

**Calculation of surface charge density**. In previous studies, the transferred charge density on the insulating surfaces was calculated by the following equation:

$$\Delta\sigma = \frac{\Delta V \varepsilon_0 \varepsilon_{sample}}{t_{sample}} \qquad (4)$$

where $\Delta\sigma$ denotes the transferred charge density, $\Delta V$ denotes the change of surface potential after the CE, $\varepsilon_0$ denotes the vacuum dielectric constant, $\varepsilon_{sample}$ denotes the relative dielectric constant of the sample and $t_{sample}$ denotes the thickness of the insulating layer.

In our experiments, the absolute charge density on the sample surface need to be calculated. In this case, the contact potential difference (CPD) between the tip and the substrate of the samples should be considered, and the absolute charge density on the insulating surfaces can be expressed as following (the calculations are shown in the Supplementary Note 5 and Supplementary Fig. 9):

$$\sigma = \frac{(V + CPD_{tip-sample})\varepsilon_0 \varepsilon_{sample}}{t_{sample}} \qquad (5)$$

where $\sigma$ denotes the absolute charge density on the sample surfaces, $V$ denotes the surface potential of the samples and the $CPD_{tip-sample}$ is the CPD between the tip and the substrate of the samples.

## Data availability

All data needed to evaluate the conclusions in the paper are present in the paper and/or the Supplementary Information. Additional data related to this paper may be requested from the authors. The source data underlying all figures can be found in the Source Data file.

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

## Acknowledgements

We would like to thank Prof. Ding Li, Prof. Xiangyu Chen, Dr. Fei Zhan, and Dr Jianjun Luo for helpful discussions. Research was supported by the National Key R & D Project from Minister of Science and Technology (2016YFA0202704), National Natural Science Foundation of China (Grant Nos. 51605033, 51432005, 5151101243, 51561145021), Beijing Municipal Science & Technology Commission (Z171100000317001, Z171100002017017, Y3993113DF).

## Author contributions

S.L. and Z.L.W. conceived the idea and designed the experiment. S.L. carried out the liquid–solid contact electrification experiments. A.C.W., S.L. and Z.L.W. contributed to the electric double-layer theory. S.L., L.X. and Z.L.W. wrote the manuscript. All the authors discussed the results and commented on the manuscript.

## Competing interests

The authors declare no competing interests.
