## [Peer Review File · Nature Communications]

Reviewers' comments:

Reviewer #1 (Remarks to the Author):

The manuscript "Quantifying electron-transfer and ion-transfer in liquid-solid contact electrification and the formation mechanism of electric double-layer" is really interesting to read and may hold an important contribution to the field. The manuscript reports the novel mechanism for liquid-solid contact electrification and separate the ion and electron transfer. However, results must be further confirmed with additional experiments and resubmitted again. The surface charge should be measured by alternative method, because the contact potential difference between the Au tip and insulator will exist even if the surface is not electrified as indicated by authors in experimental section. Also, the potential may change due to water adsorbate layers on surface and you are distinguishing not sticky charges and electrons, but chemically and physically adsorbed water. We suggest to oscillate the electrified surface against the grounded conductive plate in non-contact regime and measure the current between the ground. The surface charge then could be estimated from the current. If there is a surface charge, then there must be also a current. Also, it would be great to detect emitted electrons directly although it is very complicated in this case, because the measurements should be executed in vacuum. The method for measuring emitted electrons can be found in paper: J. Soc. Inf. Display 19, 212 (2011) 10.1889/JSID19.2.212.

Some additional suggestions to improve the quality of the manuscript:

(i) the conductivity of water used for experiments should be measured;

(ii) second sentence in the introduction claims that the recent study reveals that the dominating mechanism for triboelectrification is electron transfer. This is true for metals and semiconductors, but not for polymer insulators, for example. The different mechanisms between various materials must be distinguished and discussed in the introduction. For polymer insulators, the dominating mechanism is a heterolytic covalent bond break and material transfer as it was demonstrated in several studies from independent groups (Science 333, 308 (2011); J. Phys. Chem. C 122, 16154 (2018); Energy Environ. Sci. 12, 2417 (2019)). In fact, the ref. 1 cited by authors, where electron transfer was suggested after observing metal cation reduction on negatively charged surfaces, was proven later to be wrong, because reduction was observed also on surfaces carrying a positive net surface charge and the reactions appeared to be driven by mechanoradicals (J. Am. Chem. Soc. 134, 7223 (2012)).

(iii) the correct term for thermostimulated electron emission should be exoelectron emission. It was a popular topic to study in the 70s and 80s.

Reviewer #2 (Remarks to the Author):

This paper describes a work on liquid-solid electrification with the aim to identify the contributions of two main charging mechanisms – ion and electron transfer between the contacting surfaces. The study touches a very important fundamental scientific question related to a seemingly simple but very complex phenomenon that has many ramifications in technology. In my opinion, however, the study lacks some major technical work and the manuscript fails to provide some important experimental details that are required to justify the claims. Some of these can be listed as:

1) The authors use KPFM to measure the charge density on surfaces. This is not the ideal method to probe the (average) charge density of samples since the KPFM can only acquire data from a small region (micrometer squares) on the whole sample and the measured charge densities can vary from region to region. The patchiness of the surface potential from micro to macro scales was shown in previous work by others in references presented in this paper 2, 15, 20, and also in the following works:

Knorr N. Squeezing out hydrated protons: Low-frictional-energy triboelectric insulator charging on a microscopic scale. AIP Adv. 1, 022119 (2011).

Pandey A., Kieres J., Noras M. A. Verification of non-contacting surface electric potential measurement model using contacting electrostatic voltmeter. J. Electrostat. 67, 453-456 (2009).

Barnes A. M., Dinsmore A. D. Heterogeneity of surface potential in contact electrification under

ambient conditions: A comparison of pre- and post-contact states. J. Electrostat. 81, 76-81 (2016).

2) The reported charge densities are in $\mu\text{C}/\text{m}^2$, it is not clear how the authors ended in using these units. The experimental part does not provide clear detail on this point.

3) The very high temperatures used in the experiments might have a negative effect on KPFM measurements, in which the tip is operated very close (50 nm lift height) to the surface. At least some blank (uncharged surface) measurements should be performed to prove that the observed changes in the surface potential are not entirely or partially because of the temperature.

4) Line 108, how is the liquid vaporized? The method of vaporization can also be important in (dis)charging. Also, in the experimental part (line 356) it is stated that the samples were heated for 10 min at 513 K. Does this treatment cause any thermionic emission? (It is also stated on the same lines that the charge density was measured to be about zero except the MgO and Si₃N₄ – why is there this difference?)

5) Line 120. There is no chemical evidence that the O⁻ ions stay on the surface at these elevated temperatures. At least the surface's oxygen content should be monitored by some surface analysis, e.g. XPS, to prove the claim.

6) Water adsorption on (oxide) surfaces is well-known in surface science to be persistent to high temperatures and reduced pressures. Did authors consider any water layers and their contribution to charging? Especially the resistance of the 'sticky charges' to decay can be because of this strongly adsorbed water layer. How can the authors make sure that the surfaces are free of water? (since it is not included in the mechanism stated).

7) How the electron transfer occurs during contact electrification of the surface with water is not clear. Are there any transferable electrons in water? (I understand that the electrons are transferred from water to the surface (also as in Fig 4) since the surfaces acquire these electrons, which are later thermally emitted.)

8) Line 188. 'It is found that the charge density on the SiO₂ surfaces decreases with the increase of the NaCl concentration.' And 'Interestingly, the density of removable charges (electrons) on the SiO₂ surfaces decreases with the increase of the NaCl concentration, while the "sticky" charges (ions) density remains almost unchanged when the SiO₂ contacts with the NaCl solutions of different concentrations. It implies that the decrease of the charge density on the SiO₂ induced by the increase of the NaCl concentration is mainly due to the decrease of the electron transfer.' The authors should consider that the observation of lower charge densities with NaCl solutions might simply be because the surface is contacted with a higher dielectric constant liquid that facilitates the discharge after charging.

9) Line 209. 'It may be caused by the covering of the crystallized NaCl on the SiO₂ surface in the subsequent heating processes, which blocks the progress of ionization reaction.' Why should the ionization be blocked? As far as it is suggested in line 115, the ionization (generation of O⁻) takes place as a result of a reaction with liquid.

10) Line 267-274. 'And the hydroxyl on the solid surfaces, which is produced by the interaction of solid surface atoms with water molecules and plays an important role in the ionization reaction (chemical formula 1 and 2), is easier to generated on the hydrophilic surfaces comparing to hydrophobic surfaces. Hence, the surface ionization reaction is more likely to occur and leading to the generation of ions on the hydrophilic solid surface. On the contrary, the surface ionization reaction between the hydrophobic solid surfaces and water is less likely to occur, and the CE between the solid and liquid is electron-dominated.'

I guess this part is not correct and this statement needs revision.

11) Line 281-283. 'Furthermore, it may be possible that the electron transfer and ion transfer could be segregated on different surface areas, but remain proved experimentally.' Therefore, more data is necessary to support these claims.

12) The surface reactions ('ionizations') for other materials than SiO₂ should also be provided and discussed to make claims about the results on the mechanism of contact-charging. Each of these materials has different chemistries.

Also,

_Line 104. 'Here, flat insulating ceramic thin films, such as SiO₂, Al₂O₃, Si₃N₄ etc.' All surface names should be written here, please avoid 'etc.'

_The type of material used in contact electrification is important. Some more details should be provided on the materials. Generalizations such as 'liquid' should be avoided in the manuscript (e.g. Fig. 4, line 233-237) since the paper only describes the charging of water and aqueous solutions.

_As far as I understand Fig 4 describes only DI water case (as stated in c). In this case, the ionization should be negative. However, the surface is depicted as positive in this figure.

_ In Fig.4, 'the ions in the liquid' (as written in the caption) are shown to impact the surface. If these ions are hydroxyl ions, their concentration should be around 10⁻⁷ M at room temperature, the figure shows as if they are very abundant in the water.

Reviewer #3 (Remarks to the Author):

This manuscript presents the nature of contact electrification (CE) between liquids and solids. The electron transfer was distinguished from the ion transfer by using the thermionic emission. It is remarkable that the contribution of each transfer was investigated according to the solutes in liquid, the pH value of liquid, and the hydrophilicity of the solid. However, the reviewer thinks this manuscript needs additional data supplement to be published in Nature Communications after minor revision. The required revision data are summarized as follows.

1. The charge density of electrons decreased with the increase of the solute concentration, while the sticky charges were not affected. The manuscript explains that the absence of ionization reaction for producing the ions induces this difference. The reviewer suggest that the principle of reduced electron density should be included for comparison.

2. The detail explanations about the different ratio of electron transfer to ion transfers (E/I) are recommended for clear understanding, according to surface materials. For the MgO and the Al₂O₃, the E/I ratios are not significantly different although the water contact angle of Al₂O₃ is more than twice, compared to the MgO.

3. There is a typo.

'CE charge charges' should be revised to 'CE charges' in line 96, 4p.

4. The quality of some figures should be improved for better understanding. So below papers are recommended to be referred to enhance the informativeness of figures.

'Comprehensive biocompatibility of nontoxic and high-output flexible energy harvester using lead-free piezoceramic thin film', *Apl Materials*, 5, 074102, 2017.

'Performance-enhanced triboelectric nanogenerator enabled by wafer-scale nanogrates of multistep pattern downscaling', *Nano Energy*, 35, 415, 2017.

Quantifying electron-transfer and ion-transfer in liquid-solid contact electrification and the formation mechanism of electric double-layer

Response to Reviewers

We thank the reviewers for their constructive comments to help improve our paper. All these comments were considered in the revised manuscript. Our responses to the comments are listed below in **bold**, and the changes that were made in the revised manuscript and supporting information are marked for easy identification.

Reviewer #1

Q1. The manuscript “Quantifying electron-transfer and ion-transfer in liquid-solid contact electrification and the formation mechanism of electric double-layer” is really interesting to read and may hold an important contribution to the field. The manuscript reports the novel mechanism for liquid-solid contact electrification and separate the ion and electron transfer.

The surface charge should be measured by alternative method, because the contact potential difference between the Au tip and insulator will exist even if the surface is not electrified as indicated by authors in experimental section.

Response: Thank you very much. The Kelvin probe force microscopy (KPFM) is a common method for measuring the surface charge density on the insulator surface. Here, we added the working principle of KPFM in the Supporting Information. The relation between the surface potential (or Kelvin potential) and the surface charge density can be expressed as following (as shown in the calculations in Supporting Information):

$$V_{DC} + V_{CPD} - \frac{\sigma d_2}{\epsilon_0 \epsilon_d} = 0 \quad (\text{R1})$$

where V_{DC} denotes the DC bias applied between the tip and conductive substrate by the Kelvin controller in the system, it can also be named as ‘surface potential’ or ‘Kelvin potential’ (what we obtain in the measurement). V_{CPD} denotes the CPD between the tip and conductive substrate, σ is the charge density on the insulator surface, d_2 is the thickness of the insulator layer, ϵ_0 denotes the vacuum dielectric constant and ϵ_d denotes the relative dielectric constant of the insulator.

Indeed, the contact potential difference between the tip and insulator (Actually, it should be called as Kelvin potential here, V_{DC}) will exist, even when the surface is not electrified. When the surface is not electrified, Equation R1 can be expressed as following:

$$V_{DC} = -V_{CPD} \quad (\text{R2})$$

Equation R2 means that the Kelvin potential is opposite to the CPD between the tip and conductive substrate when the insulator surface is not electrified. In our experiments, the CPD between the tip and the conductive substrate was measured by directly scanning the substrate in KPFM mode before the substrate is covered by the insulator. Hence, the CPD between the tip and conductive substrate (V_{CPD}) is known. When the Kelvin potential is opposite to the V_{CPD} , we can know that the surface charge density of the insulator is zero. And the surface charge density can be calculated by following equation:

$$\sigma = \frac{(V_{DC} + V_{CPD})\epsilon_0\epsilon_d}{d_2} \quad (\text{R3})$$

In Equation R3, V_{CPD} , d_2 , ϵ_0 and ϵ_d are known, when the Kelvin potential (V_{DC}) is measured, the charge density on the insulator surface can be calculated quantitatively.

Q2. Also, the potential may change due to water adsorbate layers on surface and you are distinguishing not sticky charges and electrons, but chemically and physically adsorbed water.

Response: Thank you very much for the comment. The experiments were performed to demonstrate that the change of potential is not caused by the adsorbed water. The amount of the water molecules adsorbed on the insulator surface is highly dependent on the atmosphere. Here, the surface charge density of the SiO₂ (both charged and uncharged SiO₂) was measured in different atmospheres by using our home-made vacuum AFM^[R1] (50% humidity air, Ar atmosphere, 1 Pa vacuum and 0.001 Pa vacuum), as shown in Fig. S6. There were much more water molecules on the SiO₂ surface in 50% humidity air than that in 0.001 Pa vacuum condition (or that in Ar atmosphere). However, the experiment results show that no matter the SiO₂ surface was charged (Fig. S6b) or not (Fig. S6a), the surface charge density (and surface potential) remained unchanged in different atmospheres. It indicates that the adsorption and desorption of the water molecules on SiO₂ surface can not affect the surface charge density and the potential significantly.

-[R1] S. Lin, L. Xu, W. Tang, X. Chen, Z. L. Wang, *Nano Energy*. 2019, 65, 103956.

Q3. We suggest to oscillate the electrified surface against the grounded conductive plate in non-contact regime and measure the current between the ground. The surface charge then could be estimated from the current. If there is a surface charge, then there must be also a current.

Response: Thank you very much. We have considered your suggestion carefully, and it is a very good method to measure the charge density on thick insulator surface. However, we think this method is probably not suitable for measuring the surface charge density of our samples, because the thickness of the insulator layer is only 100 nm in our experiments and there is a conductive substrate below the insulator layer. In this case, the charges on the insulator surface will be shielded by the conductive substrate, and the induced charges on the grounded conductive plate and the current induced by the surface charges will not reflect the true amount of charges on the surface.

Q4. It would be great to detect emitted electrons directly although it is very complicated in this case, because the measurements should be executed in vacuum. The method for measuring emitted electrons can be found in paper: J. Soc. Inf. Display 19, 212 (2011) 10.1889/JSID19.2.212.

Response: Thank you for your suggestion. We have read the recommended paper carefully. The paper introduced a good method for direct measurement of the emission of electrons from a sample surface. However, we cannot set up the experiment in a short time, because the

measurement should be executed in vacuum, which is quite complicated for us. We will consider this method in our future works.

Q5. The conductivity of water used for experiments should be measured.

Response: The conductivity of the DI water used for experiments was measured to be 18.2 M Ω · cm.

We have added this description in the Experimental Section of the manuscript.

Q6. Second sentence in the introduction claims that the recent study reveals that the dominating mechanism for triboelectrification is electron transfer. This is true for metals and semiconductors, but not for polymer insulators, for example. The different mechanisms between various materials must be distinguished and discussed in the introduction. For polymer insulators, the dominating mechanism is a heterolytic covalent bond break and material transfer as it was demonstrated in several studies from independent groups (Science 333, 308 (2011); J. Phys. Chem. C 122, 16154 (2018); Energy Environ. Sci. 12, 2417 (2019)). In fact, the ref. 1 cited by authors, where electron transfer was suggested after observing metal cation reduction on negatively charged surfaces, was proven later to be wrong, because reduction was observed also on surfaces carrying a positive net surface charge and the reactions appeared to be driven by mechanoradicals (J. Am. Chem. Soc. 134, 7223 (2012)).

Response: As you suggested, the second sentence in the introduction was revised to “The solid-solid CE has been studied using various methods and different mechanisms were proposed (Electron transfer^[1,2], ion transfer^[3] and material transfer^[4-6] were used to explain different types of CE phenomena for various materials).” And the references were also revised accordingly.

Q7. The correct term for thermostimulated electron emission should be exoelectron emission. It was a popular topic to study in the 70s and 80s.

Response: Thank you. ‘exoelectron emission’ is perhaps the most accurate term for the thermal induced electron emission in our experiments, in which the exoelectron is emitted from insulator surface. However, considering that ‘exoelectron emission’ is not familiar to many potential readers, we prefer to use ‘thermionic emission’, which is for the general case that electron emitted from a solid surface. We believe that this is a more commonly used terminology and is familiar to many readers. Thank the referee for the suggestion.

Reviewer #2

This paper describes a work on liquid-solid electrification with the aim to identify the contributions of two main charging mechanisms – ion and electron transfer between the contacting surfaces. The study touches a very important fundamental scientific question related to a seemingly simple but very complex phenomenon that has many ramifications in technology.

Q1. The authors use KPFM to measure the charge density on surfaces. This is not the ideal method to

probe the (average) charge density of samples since the KPFM can only acquire data from a small region (micrometer squares) on the whole sample and the measured charge densities can vary from region to region. The patchiness of the surface potential from micro to macro scales was shown in previous work by others in references presented in this paper 2, 15, 20, and also in the following works:

Knorr N. Squeezing out hydrated protons: Low-frictional-energy triboelectric insulator charging on a microscopic scale. *AIP Adv.* 1, 022119 (2011).

Pandey A., Kieres J., Noras M. A. Verification of non-contacting surface electric potential measurement model using contacting electrostatic voltmeter. *J. Electrostat.* 67, 453-456 (2009).

Barnes A. M., Dinsmore A. D. Heterogeneity of surface potential in contact electrification under ambient conditions: A comparison of pre- and post-contact states. *J. Electrostat.* 81, 76-81 (2016).

Response: Thank you very much. Indeed, the scanning range of the KPFM is always limited by the scanner in auto scanning mode (93 $\mu\text{m} \times 93 \mu\text{m}$ for our AFM equipment). To overcome this limit, in our experiments, the KPFM was manually operated to scan different positions on the whole sample (more than 20 positions). In this case, the test range of the KPFM will not be limited by the scanner, and the data was acquired from the whole sample. Actually, our data also reflected the patchiness of the surface potential from micro to macro scale. As shown in Fig. 1c, the error bar of the data increases with the increasing of average value. It indicates that the charge densities on sample surface were varying from region to region when the sample was charged. But the error bar was limited even if the surface charge density is larger than $-800 \mu\text{C}/\text{m}^2$. It means that the fluctuation range of the charge density on our sample surfaces is limited.

We are sorry that this experiment detail was not mentioned in the manuscript since we have provided the error bar of the data, which means the data was statistically analysed and acquired from different positions. To address this issue, we have added this experiment detail to the Experimental Section of the manuscript.

Q2. The reported charge densities are in $\mu\text{C}/\text{m}^2$, it is not clear how the authors ended in using these units. The experimental part does not provide clear detail on this point.

Response: Thank you. We have added the detail of the calculation of the surface charge density in the Supporting Information. According to the working principle of the KPFM, the relation between the charge density on the sample surface and the surface potential (Kelvin potential) can be expressed as following:

$$\sigma = \frac{(V_{DC} + V_{CPD})\epsilon_0\epsilon_d}{d_2} \quad (\text{R4})$$

where V_{DC} denotes the DC bias applied between the tip and conductive substrate by the Kelvin controller in the system, it can be named as ‘Kelvin potential’, and it can also be named as ‘surface potential’ (what we obtain in the measurement). V_{CPD} denotes the contact potential difference between the tip and conductive substrate, σ is the charge density on the insulator surface, d_2 is the thickness of the insulator layer, ϵ_0 denotes the vacuum dielectric constant and ϵ_d denotes the relative dielectric constant.

As an example, the surface charge density of a SiO₂ sample after contacting with DI water was calculated here. The surface potential (Kelvin potential, V_{DC}) of the SiO₂ sample after contacting with DI water was measured to be -2550 mV, V_{CPD} was directly measured to be 210 mV by scanning the substrate in KPFM mode, $d_2 = 100 \text{ nm}$, $\epsilon_0 = 8.85 \times 10^{-12} \text{ F/m}$, and $\epsilon_{SiO_2} = 3.9$, Hence, Equation R4 can be expressed as following:

$$\begin{aligned}\sigma &= \frac{(-2.55 \text{ V} + 0.21 \text{ V}) \times 8.85 \times 10^{-12} \frac{\text{F}}{\text{m}} \times 3.9}{100 \text{ nm}} = -80.76 \times 10^{-14} \frac{\text{VF}}{\text{nm} \cdot \text{m}} \\ &= -80.76 \times 10^{-5} \frac{\text{C}}{\text{m}^2} = -807.6 \frac{\mu\text{C}}{\text{m}^2}\end{aligned}\quad (\text{R5})$$

According to Equation R5, the unit of the surface charge density is in $(\frac{\mu\text{C}}{\text{m}^2})$.

Q3. The very high temperatures used in the experiments might have a negative effect on KPFM measurements, in which the tip is operated very close (50 nm lift height) to the surface. At least some blank (uncharged surface) measurements should be performed to prove that the observed changes in the surface potential are not entirely or partially because of the temperature.

Response: Thank you very much for your suggestion. We have performed the blank measurements, as shown in Fig. S7. The charge density of the uncharged SiO₂ surfaces was measured at different temperature for 40 mins (393 K, Fig. S7a; 473 K, Fig. S7b). The results show that the charge density of the SiO₂ surfaces remained zero at 393 K and 473 K. It indicates that the observed changes in the surface potential in our experiments are not due to the temperature effects on the measurements.

Q4. Line 108, how is the liquid vaporized? The method of vaporization can also be important in (dis)charging. Also, in the experimental part (line 356) it is stated that the samples were heated for 10 min at 513 K. Does this treatment cause any thermionic emission? (It is also stated on the same lines that the charge density was measured to be about zero except the MgO and Si₃N₄ – why is there this difference?)

Response: In our experiments, the liquid was vaporized by blowing with dry Ar, and then, the sample was tested in an Ar atmosphere.

All the samples were heated for 10 min at 513 K before the experiments; this treatment will lead to the thermionic emission. The purpose of this treatment is to remove the initial charges on the sample surface, which may affect the charge transfer in the experiments. (Some electrons may be generated on the sample surface during the sample preparation.)

The charge density on the MgO and Si₃N₄ surfaces were not zero after the heating treatment. This may be attributed to that the MgO and Si₃N₄ are very hydrophilic, the ions (which cannot be removed by heating at 513 K) will be generated by the ionization reaction on the MgO and Si₃N₄ surfaces, once they are exposed to humid air.

Q5. Line 120. There is no chemical evidence that the O⁻ ions stay on the surface at these elevated temperatures. At least the surface's oxygen content should be monitored by some surface analysis, e.g.

XPS, to prove the claim.

Response: Thank you very much. We performed the *ab initio* molecular dynamics to simulate the behaviour of the O^- ion on the sample surfaces at 513 K, which was the highest temperature used in the experiments (the *ab initio* molecular dynamics simulations have been added to the Supporting Information). The simulation results suggest that O^- ion will stay and oscillate on the sample surfaces at 513 K. The calculation details and the results (the videos) are shown in Supporting Information.

The number ratio of O to Si on SiO_2 surface was measured by using XPS, and no significant difference was found between charged and uncharged SiO_2 . This is because there is only $\sim 1 O^-$ ion per 1000 nm^2 on the charged SiO_2 , as we calculated in page 12. The change of the number ratio of O to Si is negligible after charging.

(The number ratio of O to Si on uncharged SiO_2 surface was measured to be 2.043; The number ratio of O to Si on charged SiO_2 surface was measured to be 2.052. The change in the number ratio of O to Si was too small to prove the O atom on the surface increased after charging.)

Q6. Water adsorption on (oxide) surfaces is well-known in surface science to be persistent to high temperatures and reduced pressures. Did authors consider any water layers and their contribution to charging? Especially the resistance of the 'sticky charges' to decay can be because of this strongly adsorbed water layer. How can the authors make sure that the surfaces are free of water? (Since it is not included in the mechanism stated).

Response: Thank you very much.

We cannot make sure that the surfaces are free of water, because the water adsorption on solid surfaces was found to be persistent to high temperatures and even ultra-high vacuum.

Here, we added some experiments in the Supporting Information to demonstrate that the water layers have no significant contributions to the surface charge density in our experiments. As we know, the amount of water molecules adsorbed on the insulator surface is highly dependent on the atmosphere. The surface charge density of the SiO_2 (both charged and uncharged SiO_2) was measured in different atmospheres by using our home-made vacuum AFM^[R2] (50% humidity air, Ar atmosphere, 1 Pa vacuum and 0.001 Pa vacuum), as shown in Fig. S6. There were much more water molecules on the SiO_2 surface in 50% humidity air than that in 0.001 Pa vacuum condition (or that in Ar atmosphere). However, the results show that no matter the SiO_2 surface was charged (Fig. S6b) or not (Fig. S6a), the surface charge density remained unchanged in different atmospheres. It indicates that the adsorption and desorption of the water molecules on SiO_2 surface can not affect the surface charge density significantly.

-[R2] S. Lin, L. Xu, W. Tang, X. Chen, Z. L. Wang, *Nano Energy*. 2019, 65, 103956.

Q7. How the electron transfer occurs during contact electrification of the surface with water is not clear. Are there any transferable electrons in water? (I understand that the electrons are transferred from water to the surface (also as in Fig 4) since the surfaces acquire these electrons, which are later thermally emitted.)

Response: The quantum mechanical calculations suggested that the electrons may transfer between two atoms once the electron clouds of two atoms overlap with each other.^[R3] And the electron transfer between two materials has been described in “electron-cloud-potential-well model”, which was proposed in our previous studies.^[R4] According to the model, the electron transfer between the surface and water can be described clear. When the liquid contacts a solid surface, the water molecules will impact the solid surface due to the thermal motion and the pressure from the liquid. During the impact, the electron clouds of the solid surface atoms and water molecules may overlap with each other, and lead to electron transfer between the solid atoms and water molecules.

“electrons will transfer between the solid atoms and water molecules owing to the interatomic interaction.” in page 10 was revised to “electrons will transfer between the solid atoms and water molecules owing to the overlap of the electron clouds of the solid atoms and water molecules in the impact.” And relevant references are quoted in the manuscript.

-[R3] M. Willatzen, Z. L. Wang, *Nano Energy* 2018, 52, 517-523.

-[R4] C. Xu, Y. Zi, A. Wang, H. Zou, Y. Dai, X. He, P. Wang, Y. Wang, P. Feng, D. Li, Z. L. Wang, *Adv. Mater.* 2018, 30, 1706790.

Q8. Line 188. ‘It is found that the charge density on the SiO₂ surfaces decreases with the increase of the NaCl concentration.’ And ‘Interestingly, the density of removable charges (electrons) on the SiO₂ surfaces decreases with the increase of the NaCl concentration, while the “sticky” charges (ions) density remains almost unchanged when the SiO₂ contacts with the NaCl solutions of different concentrations. It implies that the decrease of the charge density on the SiO₂ induced by the increase of the NaCl concentration is mainly due to the decrease of the electron transfer.’

The authors should consider that the observation of lower charge densities with NaCl solutions might simply be because the surface is contacted with a higher dielectric constant liquid that facilitates the discharge after charging.

Response: Thank you very much for you reminding. The focus of this paper was to distinguish ion and electron on a charged surface. We determined that NaCl concentration affected the electron transfer instead of the ion transfer, but forgot to give an explanation for the decrease in electron transfer. The discharge of the electrons is a very possible reason for the decrease of the electron transfer. As you suggested, we added this explanation to the manuscript.

Q9. Line 209. ‘It may be caused by the covering of the crystallized NaCl on the SiO₂ surface in the subsequent heating processes, which blocks the progress of ionization reaction.’ Why should the ionization be blocked? As far as it is suggested in line 115, the ionization (generation of O⁻) takes place as a result of a reaction with liquid.

Response: This was only happened in the charging and heating cycle tests. When the water on the SiO₂ surface was vaporized after the contacting with NaCl solution, some NaCl may crystallize and cover the SiO₂ surface. In subsequent contact between the SiO₂ and NaCl solution, some places on the SiO₂ surface were cover by the NaCl (the NaCl needed some time to

be dissolved), and the water molecules in NaCl solution cannot contact the SiO₂ surface. Hence, the ionization reaction will not happen in these places.

Q10. Line 267-274. 'And the hydroxyl on the solid surfaces, which is produced by the interaction of solid surface atoms with water molecules and plays an important role in the ionization reaction (chemical formula 1 and 2), is easier to generated on the hydrophilic surfaces comparing to hydrophobic surfaces. Hence, the surface ionization reaction is more likely to occur and leading to the generation of ions on the hydrophilic solid surface. On the contrary, the surface ionization reaction between the hydrophobic solid surfaces and water is less likely to occur, and the CE between the solid and liquid is electron-dominated.'

I guess this part is not correct and this statement needs revision.

Response: We revised the statement to “And the Oxygen atoms or Hydrogen atoms in water molecules are more likely to form covalent bonds with the atoms on the hydrophilic surface. In other words, the surface ionization reaction is more likely to occur and leading to the generation of ions on the hydrophilic solid surface. On the contrary, the surface ionization reaction between the hydrophobic solid surfaces and water is less likely to occur, and the CE between the solid and aqueous is electron-dominated.” As shown in line 267-274 in the manuscript.

Q11. Line 281-283. 'Furthermore, it may be possible that the electron transfer and ion transfer could be segregated on different surface areas, but remain proved experimentally.' Therefore, more data is necessary to support these claims.

Response: The experiments showed that the solid surface can receive both electrons and positive ions at the same time. We suggested that the electrons and positive ions should be segregated on different surface areas, because the average distance between charge distribution on surfaces is very spare, and the average distance between two adjacent charges is ~30-100 nm. Otherwise, the electrons and positive ions will neutralize each other. If the density gets higher, surface discharge would occur due to electric breakdown. The surface potential of the charged surface measured by using our KPFM equipment was quite uniform (because its resolution is limited). We think this claim can only be proved by the KPFM with atomic-scale contrast.^[R5]

We agree with the reviewer that more data is necessary to support this hypothesis and it is very interesting and important to prove this statement. We will discuss this topic in our future studies.

-[R5] L. Nony, A. Foster, F. Bocquet, C. Loppacher, *Phy. Rev. Lett.* 2009, 103, 036802.

Q12. The surface reactions ('ionizations') for other materials than SiO₂ should also be provided and discussed to make claims about the results on the mechanism of contact-charging. Each of these materials has different chemistries.

Response: Thank you very much. The description of the ionization reactions between the water and the materials used in this work has been added to the Supporting Information.

As described in previous studies,^[R6] the ionization reaction between oxide surface and

water is caused by the amphoteric surface groups, and the ionization reaction can be expressed as following:

where ‘A’ represents the oxidized atom, such as ‘Si’ in SiO₂, ‘Al’ in Al₂O₃.^[R7]

For Si₃N₄, there are amine groups (SiNH₂) on the surface^[R8] and adsorb the H⁺ ions as shown below^[R9]:

For AlN, it has been demonstrated that water molecules will induce N vacancy on the AlN surface, and the N vacancy will be occupied by the O²⁻.^[R10] And then, the ionization reactions (R6) and (R7) will occur since there are Al-O bonds on the AlN surface.

-[R6] S. Usui, J. Coll. Inter. Sci. 2008, 320, 353-359.

-[R7] L. Bousse, N. Rooij, P. Bergveld, IEEE T. Electron Dev. 1983, ED-30, 1263-1270.

-[R8] R. Raiteri, S. Martinoia, M. Grattarola, Biosens. Bioelectron. 1995, 11, 1009-1017.

-[R9] C. Galassi, F. Bertoni, J. Mater. Res. 2000, 15, 155-163.

-[R10] Y. Chen, X. Hou, Z. Fang, E. Wang, J. Chen, G. Bei, J. Phys. Chem. C 2019, 123, 5460-5468.

Q13. Line 104. ‘Here, flat insulating ceramic thin films, such as SiO₂, Al₂O₃, Si₃N₄ etc.’ All surface names should be written here, please avoid ‘etc.’.

Response: Thank you very much. we have revised the manuscript as you suggested.

Q14. The type of material used in contact electrification is important. Some more details should be provided on the materials. Generalizations such as ‘liquid’ should be avoided in the manuscript (e.g. Fig. 4, line 233-237) since the paper only describes the charging of water and aqueous solutions.

Response: Thank you for your reminding. We have revised the manuscript as you suggested.

Q15. As far as I understand Fig 4 describes only DI water case (as stated in c). In this case, the ionization should be negative. However, the surface is depicted as positive in this figure.

Response: Thank you. In fact, the surface is depicted as negative in the figures. As shown in Fig. 4c, there are both anions and electrons on the charged surface. Hence, the surface is negatively charged.

Q16. In Fig.4, ‘the ions in the liquid’ (as written in the caption) are shown to impact the surface. If these ions are hydroxyl ions, their concentration should be around 10^{-7} M at room temperature, the figure shows as if they are very abundant in the water.

Response: Thank you for your reminding. Here, the ions are not just hydroxyl ions, but all the ions in the solution.

Reviewer #3

This manuscript presents the nature of contact electrification (CE) between liquids and solids. The electron transfer was distinguished from the ion transfer by using the thermionic emission. It is remarkable that the contribution of each transfer was investigated according to the solutes in liquid, the pH value of liquid, and the hydrophilicity of the solid. However, the reviewer thinks this manuscript needs additional data supplement to be published in Nature Communications after minor revision. The required revision data are summarized as follows.

Q1. The charge density of electrons decreased with the increase of the solute concentration, while the sticky charges were not affected. The manuscript explains that the absence of ionization reaction for producing the ions induces this difference. The reviewer suggest that the principle of reduced electron density should be included for comparison.

Response: Thank you very much for your suggestion. In the manuscript, the effect of NaCl concentration on the charge transfer was explained as you suggested (“It implies that the decrease of the charge density on the SiO₂ induced by the increase of the NaCl concentration is mainly due to the decrease of the electron transfer” in page 7). The absence of ionization reaction was used to explain the difference of saturated ion density on the SiO₂ surface in the charging and heating cycle tests. (Fig. 2c and Fig. 1d)

In the charging and heating cycle tests, when the water on the SiO₂ surface was vaporized after contacting with NaCl solution, some NaCl may crystallize and cover the SiO₂ surface. In subsequent contact between the SiO₂ and NaCl solution, some places on the SiO₂ surface were cover by the NaCl (the NaCl needed some time to be dissolved), and the water molecules in NaCl solution cannot contact the SiO₂ surface. Hence, the ionization reaction will not happen in these places.

Q2. The detail explanations about the different ratio of electron transfer to ion transfers (E/I) are recommended for clear understanding, according to surface materials. For the MgO and the Al₂O₃, the E/I ratios are not significantly different although the water contact angle of Al₂O₃ is more than twice, compared to the MgO.

Response: Thank you for your reminding. According to the experiment results, we believe that the E/I ratio is highly affected by the hydrophilicity of the solid surface, as described in the manuscript (in page 9). The data show that the E/I ratio increased significantly only when the water contact angle (WCA) of the solid sample is greater than 90 °. For the MgO and the Al₂O₃,

their WCA are smaller than 90 °, and the E/I ratios of them were not significantly different. It needs more data for further understanding about these results, and we will discuss this topic in our future studies.

Q3. There is a typo.

‘CE charge charges’ should be revised to ‘CE charges’ in line 96, 4p.

Response: Thank you for your reminding. We have revised the manuscript as you suggested.

Q4. The quality of some figures should be improved for better understanding. So below papers are recommended to be referred to enhance the informativeness of figures.

‘Comprehensive biocompatibility of nontoxic and high-output flexible energy harvester using lead-free piezoceramic thin film’, *Apl Materials*, 5, 074102, 2017.

‘Performance-enhanced triboelectric nanogenerator enabled by wafer-scale nanogrates of multistep pattern downscaling’, *Nano Energy*, 35, 415, 2017.

Response: Thank you very much for your recommendations. We have revised our figures accordingly (the E/I ratios were added in Fig. 2a).

REVIEWERS' COMMENTS:

Reviewer #1 (Remarks to the Author):

The comments have been addressed. I am satisfied with most of the answers. However, the manuscript can't be accepted for publication without providing clear evidence for the reported phenomenon. In particular, the unchanged surface charge density in different atmospheres does not indicate that during the heating the differences in measured Kelvin potential are caused by desorption of physically and/or chemically adsorbed water. The reported surface charge values are very large and thus the current against the ground from the oscillating conductive plate in a non-contact regime should be observable (if there is any charge). If the insulative sample side will be exposed to the oscillating plate, the electric field will not be shielded. It may be so that this measurement will not reflect the very true amount of surface charges, but it will prove without a doubt that there is an actual surface charge.

Reviewer #2 (Remarks to the Author):

In the revised manuscript, the authors addressed all the points that were unclear in the previous version. Although I believe the 'thermionic emission' phenomenon has to be elaborated with more experiments and supported with detailed surface chemical analysis, one should admit that it will not be possible in one report. I hope that this study serves as a starting point for the following detailed understanding of the phenomenon.

Reviewer #3 (Remarks to the Author):

The revised manuscript presents the more informative explanation about nature of contact electrification (CE) between liquids and solids, and the difference between electron transfer and ion transfer. It is noteworthy that the detail investigation of contribution was clearly described according to the solutes in liquid, the pH value of liquid, and the hydrophilicity of the solid. The reviewer thinks the revised manuscript includes enough modifications for comments.

Quantifying electron-transfer in liquid-solid contact electrification and the formation of electric double-layer

Response to Reviewers

We thank the reviewers for their constructive comments again. Our responses to the comments are listed below in **bold**.

Reviewer #1

Q1: The comments have been addressed. I am satisfied with most of the answers. However, the manuscript can't be accepted for publication without providing clear evidence for the reported phenomenon. In particular, the unchanged surface charge density in different atmospheres does not indicate that during the heating the differences in measured Kelvin potential are caused by desorption of physically and/or chemically adsorbed water.

Response: Thank you very much for your comments. You are right, the unchanged surface charge density in different atmospheres does not indicate that during the heating the differences in measured Kelvin potential are caused by desorption of physically and/or chemically adsorbed water.

In the manuscript, we did not claim that the changes of Kelvin potential during the heating are caused by desorption of physically and/or chemically adsorbed water. On the contrary, what we wanted to prove was that the changes of Kelvin potential were caused by the thermionic emission of electrons instead of the desorption of physically and/or chemically adsorbed water during the heating. (As described in Supplementary Note 3. "It indicates that the adsorption and desorption of the water molecules on SiO₂ surface can not affect the surface charge density significantly.")

Q2: The reported surface charge values are very large and thus the current against the ground from the oscillating conductive plate in a non-contact regime should be observable (if there is any charge). If the insulative sample side will be exposed to the oscillating plate, the electric field will not be shielded. It may be so that this measurement will not reflect the very true amount of surface charges, but it will prove without a doubt that there is an actual surface charge.

Response: Thank you very much. As you suggested, the current against the ground from the conductive plate (copper) when the charged SiO₂ sample oscillating in a non-contact regime was observed, which means that there was an actual charge on the SiO₂ surface after contact with DI water. The experiments and results are shown in Fig. R1. As shown in Fig. R1a, the voltage and the current from conductive plate against ground is measured by using the Keithley 6514, and the SiO₂ sample is driven by a linear motor to oscillate in front of the conductive plate (in a non-contact regime). The pictures of the conductive plate and the SiO₂ sample are shown in Fig. R1b and c, respectively, and the experiment is shown in Supplementary Movie 8. When the SiO₂ was uncharged (before contact with DI water), the voltage of the conductive plate is shown in Fig. R1d, it can be seen that the change of the voltage was very limited. When the SiO₂ was

charged (after contact with DI water), the change of the voltage of the conductive plate can be observed (Fig. R1e), it means that the charged SiO_2 sample can induce the charges on the conductive plate surface. Further, the current from the conductive plate against ground was measured when the charged SiO_2 sample (after the contact with DI water) oscillating in front of the conductive plate, as shown in Fig. R1f. The results show that the SiO_2 sample after contact with DI water can induced the current from conductive plate against ground, which suggests there is an actual charge on the SiO_2 sample surface after the contact with DI water.

Figure R1. The experiments to prove there is an actual charge on the SiO_2 surface. (a) The setup of the experiments. (b) The picture of the conductive plate in the experiment. (c) The picture of the SiO_2 sample. (d) The voltage of the conductive plate against ground when the uncharged SiO_2 sample oscillating in front of the conductive plate. (e) The voltage of the conductive plate against ground when the charged SiO_2 sample (after the contact with DI water) oscillating in front of the conductive plate. (f) The current from the conductive plate against ground when the charged SiO_2 sample (after the contact with DI water) oscillating in front of the conductive plate.

Reviewer #2

In the revised manuscript, the authors addressed all the points that were unclear in the previous version. Although I believe the 'thermionic emission' phenomenon has to be elaborated with more experiments and supported with detailed surface chemical analysis, one should admit that it will not be possible in one report. I hope that this study serves as a starting point for the following detailed understanding of the phenomenon.

Response: Thank you very much for your comment. We will do more experiments and surface chemical analysis to support the 'thermionic emission' phenomenon in our future studies.

Reviewer #3

The revised manuscript presents the more informative explanation about nature of contact electrification (CE) between liquids and solids, and the difference between electron transfer and ion transfer. It is noteworthy that the detail investigation of contribution was clearly described according to the solutes in liquid, the pH value of liquid, and the hydrophilicity of the solid. The reviewer thinks the revised manuscript includes enough modifications for comments.Q1. The charge density of electrons decreased with the increase of the solute concentration, while the sticky charges were not affected. The manuscript explains that the absence of ionization reaction for producing the ions induces this difference. The reviewer suggest that the principle of reduced electron density should be included for comparison.

Response: Thank you very much for your comment. As you suggested, the principle of reduced electron density was included for comparison in the manuscript (page 6, first paragraph).